# Light-dependent grazing can drive formation and deepening of deep chlorophyll maxima

Holly V. Moeller [1], Charlotte Laufkötter [2,3], Edward M. Sweeney[4,5] & Matthew D. Johnson[6]

Deep Chlorophyll Maxima (DCMs) are subsurface peaks in chlorophyll-*a* concentration that may coincide with peaks in phytoplankton abundance and primary productivity. Work on the mechanisms underlying DCM formation has historically focused on phytoplankton physiology (e.g., photoacclimation) and behavior (e.g., taxis). While these mechanisms can drive DCM formation, they do not account for top-down controls such as predation by grazers. Here, we propose a new mechanism for DCM formation: Light-dependent grazing by microzooplankton reduces phytoplankton biomass near the surface but allows accumulation at depth. Using mathematical models informed by grazing studies, we demonstrate that light-dependent grazing is sufficient to drive DCM formation. Further, when acting in concert with other mechanisms, light-dependent grazing deepens the DCM, improving the fit of a global model with observational data. Our findings thus reveal another mechanism by which micro-zooplankton may regulate primary production, and impact our understanding of biogeo-chemical cycling at and above the DCM.

[1] Department of Ecology, Evolution and Marine Biology, University of California, Santa Barbara, Santa Barbara, CA 93106-9620, USA. [2] Climate and Environmental Physics, Physics Institute, University of Bern, Hochschulstrasse 6, 3012 Bern, Switzerland. [3] Program in Atmospheric and Oceanic Sciences, Princeton University, Princeton 08540 NJ, USA. [4] Sea Education Association, 171 Woods Hole Road, Falmouth, MA 02540, USA. [5] Education Department, Santa Barbara Museum of Natural History, Santa Barbara 93105 CA, USA. [6] Biology Department, Woods Hole Oceanographic Institution, 266 Woods Hole Road, Woods Hole, MA 02543, USA. Correspondence and requests for materials should be addressed to H.V.M. (email: holly.moeller@lifesci.ucsb.edu)

D eep chlorophyll maxima (DCMs), subsurface peaks in the concentration of the photosynthetic pigment chlorophyll-*a* (chl-*a*), are widespread phenomena observed in aquatic ecosystems, from freshwater lakes to coastal oceans to oligo-trophic gyres[1–3]. Particularly in stratified systems, DCMs may form and persist over weeks to months. While DCMs are not always coincident with biomass or productivity maxima[4,5], they often represent peaks in abundance of a unique phytoplankton community (i.e., differing substantially in species membership from surface waters[6–9]). Furthermore, photosynthesis occurring at the DCM may result in substantial contributions to local pri-mary production[10,11] and, because of its relative proximity to export depth thresholds, may be an important part of the bio-logical pump[12,13].

A number of hypotheses have been proposed for DCM for-mation and maintenance. These hypotheses largely focus on bottom-up controls: limitations of phytoplankton abundance and productivity based on interactions between these primary pro-ducers and their abiotic environment[1]. First, phytoplankton are fundamentally limited by the availability of light. In low-light environments, phytoplankton compensate by increasing their per-cell photosynthetic pigment production[14,15]. Because light attenuates with depth, this photoacclimation mechanism can produce a DCM[16,17], though it cannot explain deep biomass maxima. Phytoplankton also experience a tradeoff between light, which is supplied by solar irradiance from the surface of the water column, and nutrients, which are supplied by upwelling and diffusion from below. As nutrients become depleted from surface waters, phytoplankton accumulate at progressively deeper depths until light becomes co-limiting[18,19]. This co-limitation mechan-ism is supported by DCMs that coincide with the nutricline[20] and that deepen as light increases or nutrients decrease[21,22]. Finally, taxis[23,24] and buoyancy regulation[25,26] by phytoplankton can result in behavioral mechanisms producing narrow, subsurface biomass layers. Particularly following blooms, sinking of nutrient-stressed cells may also result in a transient DCM[27,28].

However, top-down controls (e.g., removal by higher trophic levels) are also known to be important drivers of phytoplankton abundance. For example, grazers can regulate phytoplankton abundance and composition[29,30], viruses may drive termination of blooms[31,32], and in several Earth System Models grazers can even cause a decrease in future primary production[33]. Yet, such mechanisms have generally been disregarded in DCM formation, perhaps because we expect an individual grazer's functional traits (e.g., per-capita ingestion rate) to be independent of depth. One exception has been the observation that mixotrophs (here defined as organisms that simultaneously engage in phototrophy and phagotrophic heterotrophy) may cause DCM formation when they are weaker competitors for light than their phytoplankton prey and thus accumulate and exert top-down control on prey populations at the surface. If the mixotrophs contain relatively low amounts of chlorophyll compared to the phytoplankton, the DCM would then coincide with a peak in phytoplankton prey biomass at depth[34]. In this case, however, it is the photosynthetic traits of the organisms, rather than those associated with het-erotrophy, that are invoked to explain DCM formation.

Thus, though distributions of grazers and phytoplankton, and therefore total rates of removal by grazing, may vary with depth, we typically view this as a consequence of phototroph distribution (i.e., regulated by bottom-up mechanisms), as opposed to a result of differential grazing (i.e., via a top-down mechanism). Here, we challenge this conventional view by proposing an alternate mechanism for DCM formation: light-dependent grazing of phytoplankton by microzooplankton can drive the formation, and increase the depth, of DCMs due to elevated rates of removal of phytoplankton in shallower waters.

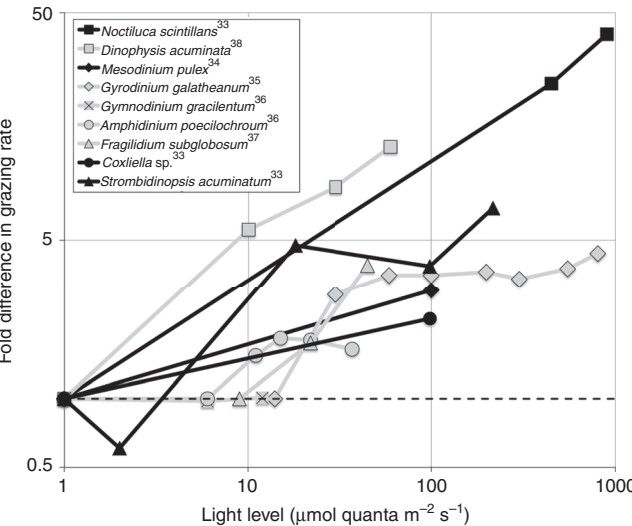

**Fig. 1** Laboratory measurements of grazing rates. Data are presented for microzooplankton (black lines and symbols) and mixotrophs (gray lines and symbols). For comparison, all rates have been normalized to grazing rates in darkness (light level = 0). The dashed line at fold difference = 1 indicates the expected relative grazing rate when grazing is not affected by light. In all but one of the studies surveyed, grazing rates increased with increasing light availability. Note log scale of both axes

Using controlled laboratory experiments, Suzanne Strom (2001)[35] was the first to report that microzooplankton ingestion and digestion rates of pigmented prey (i.e., phytoplankton) increase with increasing availability of light. Subsequently, a number of other studies have reported a positive relationship between light and grazing (Fig. 1) in heterotrophs[36] and mixo-trophs[37–40] (here defined as protists that combine photosynthesis and phagotrophic heterotrophy for growth).

The mechanism underlying the light dependence of grazing is not fully known. One possible explanation is that the process of breaking down chlorophyll may require the production of reac-tive oxygen species from the pigments themselves[35,41], so light absorption by prey contained within a digestive vacuole may increase digestion rates. Second, as unicellular organisms, microzooplankton risk photooxidative stress when grazing on pigmented prey because their digestive vacuoles are in close proximity to other critical cellular machinery. Chlorophyll is known to produce singlet oxygen radicals[42], and numerous protist grazers have the capacity to detoxify chlorophyll to the catabolite $13^2,17^3$-cyclopheophorbide *a* enol, which accumulates in cells and may act as an antioxidant[43]. Thus, more rapid digestion may be an adaptive response to reduce cell damage in high-light environments[35].

Both light-aided digestion and photooxidative stress should particularly affect small-bodied, translucent grazers (e.g., micro-zooplankton). In a water column, where light attenuates with depth, if a grazer's consumption rate depends on light availability, its grazing potential will be highest at the surface and decline with depth. Thus, in deeper, darker layers of the water column, phy-toplankton may have a spatial refuge where predation pressure is alleviated, allowing accumulation and formation of a DCM. While this possibility has occasionally been alluded to in the literature[1,8], it seems to have so far been dismissed without thorough investigation.

Here, we use two modeling approaches—a one-dimensional (1-D) model representing the focal mechanism and a three-dimensional (3-D) marine ecosystem model—to argue that, in fact, this is a viable mechanism for DCM formation.

## Results

**Light-dependent grazing is sufficient to drive DCM formation.**
To test our hypothesis that light-dependent grazing can drive DCM formation, we developed a 1-D model for a water column in which we account for the availability of light (the sole limiting resource, formulated after[44]) and the population dynamics of phytoplankton and microzooplankton.

Our model deliberately incorporates relatively few processes: first, surface irradiance is fixed at a constant input rate, $I_{in}$. Light availability decays with depth following the Lambert–Beer Law as it is absorbed by water (with a baseline absorptivity $k_0$), phytoplankton (with a per-biomass absorptivity $k_W$; note that by treating this parameter as a constant we assume no photoacclimation), and microzooplankton (absorptivity $k_Z$). Thus, the in situ light availability at any focal depth $z$ can be found by integrating absorption in the water column above the focal depth (represented by the integral over the spatial coordinate, $s$):

$$I(z) = I_{in} \exp\left[-\int_0^z k_0 + k_P P(s) + k_Z Z(s) \mathrm{d}s\right]. \quad (1)$$

Second, phytoplankton $P$ growth is light limited: photosynthesis rates are an increasing saturating function of in situ light with a maximum rate $p$ and a half-saturation irradiance $H_P$. We do not model nutrient dynamics in this model formulation, in order to highlight the sufficiency of the light-dependent grazing mechanism. Phytoplankton respiratory loss rates $l$ are constant, and phytoplankton may be consumed by grazers. These microzooplankton $Z$ feed on phytoplankton with a maximum per-capita grazing rate $g$ that is an increasing, saturating function of light availability (half-saturation irradiance $H_Z$). The microzooplankton also follow the commonly observed Holling Type II functional response[45], so per-capita grazing rate is also a saturating function of prey abundance (half-saturation prey abundance $H_A$). Preys are turned into predator biomass with a conversion efficiency $e$, and microzooplankton have a mortality rate $m$. Finally, we consider all organisms to be neutrally buoyant and to move through diffusion $D$. Collectively, this gives rise to two partial differential equations describing the local population dynamics of the phytoplankton and microzooplankton:

$$\frac{\partial P}{\partial t} = P(z)\left[\frac{pI(z)}{H_P + I(z)} - l - \frac{gI(z)}{H_Z + I(z)} \cdot \frac{Z(z)}{H_A + P(z)}\right] + D\frac{\partial^2 P(z)}{\partial z^2}, \quad (2)$$

$$\frac{\partial Z}{\partial t} = Z(z)\left[\frac{egI(z)}{H_Z + I(z)} \cdot \frac{P(z)}{H_A + P(z)} - m\right] + D\frac{\partial^2 Z(z)}{\partial z^2}. \quad (3)$$

Our 1-D model demonstrates that light-dependent grazing alone is a sufficient mechanism to generate DCMs: within a broad range of parameter values (surface light levels sufficiently high to support phytoplankton and zooplankton populations, and diffusion rates that are slow relative to photosynthesis and grazing rates such that the water column is semi-stratified on the timescale of organism lifespan), subsurface phytoplankton biomass maxima develop (Fig. 2, Supplementary Fig. 1–2). Photoacclimation is absent, so this corresponds to a DCM. Above the DCM, phytoplankton are limited by top-down grazing; below the DCM, phytoplankton are limited by low light levels (Fig. 2).

These DCMs deepen with increasing surface light, which is also consistent with observations of DCM seasonality in four oceanic regions[21]. Although increasing surface light expands the region of the water column where phytoplankton can grow, it also alleviates light limitation of grazing. Thus, our model predicts that phytoplankton biomass at the DCM should hold roughly constant, while microzooplankton biomass increases with

increasing surface light (Fig. 2d, e). This contrasts with the prediction of the nutrient/light co-limitation mechanism that DCM depth and chl-*a* content (from both photoacclimation and increasing phytoplankton biomass) should increase with increasing light[21]. These predictions appear to be a general consequence of incorporating light-dependent microzooplankton grazing: other model formulations, including a Holling Type I functional response, a linear (rather than saturating) relationship between light availability and grazing rate, and a formulation in which prey handling time (rather than grazing rate) was a function of light all produced qualitatively identical results (Supplementary Fig. 3).

**Global deepening of the DCM improves model-data agreement.** We next evaluated the extent to which DCM depth at the global scale is impacted by light-dependent grazing. We incorporated light-dependent grazing into the Carbon, Ocean Biogeochemistry and Lower Trophics (COBALT) global marine biogeochemistry model[46], which is coupled to the Earth System Model ESM2M developed at the Geophysical Fluid Dynamic Laboratory[47,48]. COBALT explicitly models multiple phytoplankton and zooplankton functional classes[46], making it ideally suited for the incorporation of light-dependent grazing. Further, COBALT already accounts for photoacclimation and nutrient-light co-limitation; thus, addition of light-dependent grazing allows us to determine how this mechanism interacts with other recognized mechanisms for DCM formation.

We modified the grazing rate of the COBALT microzooplankton class (body size <200 μm[49]) such that grazing was a function of light. COBALT parameterizes grazing with a Type II functional response:[45] that is, per-zooplankter grazing rates are a saturating function of prey density[46]. Therefore, we specifically modified the maximum achievable grazing rate (when preys are in excess) to be a function of local light availability. Microzooplankton were assigned a baseline grazing rate $g_0$, which could be increased by a maximum of $g_1$ as a saturating function of light availability:

$$g_{mz} = g_0 + g_1\left[\frac{I}{H_{mz} + I}\right], \quad (4)$$

where $H_{mz}$ is the irradiance at which grazing rates have increased by half of $g_1$ (i.e., $g_{mz} = g_0 + g_1/2$).

Modifying the grazing function in a complex marine ecosystem model with several interacting phyto- and zooplankton types results in a disturbance of the carefully equilibrated food web. We therefore tested parameters from a range of possible values within the weak experimental constraints to determine a parameter set that improved the match between the model's predicted DCM depths and observed data, while keeping phyto- and zooplankton biomass within the observed range. Specifically, we set $g_0$ to 0.15 prey pred$^{-1}$ d$^{-1}$, with a maximum increase $g_1$ of an additional 1.65 prey pred$^{-1}$ d$^{-1}$. We set the half-saturation irradiance $H_{mz}$ at 9 W m$^{-2}$ (~40 μmol quanta m$^{-2}$ s$^{-1}$), a relatively low irradiance compared to surface light levels which range to >2000 μmol quanta m$^{-2}$ s$^{-1}$ during midday in summer. This estimate also falls within the range demonstrated by laboratory experiments (Fig. 1) and is conservative in that it allows microzooplankton to achieve maximum grazing rates at relatively low light levels. The model was then spun up for 20 years after the modification of the grazing function to allow the plankton community to reach a new equilibrium.

We find that this modification results in an overall deepening of the DCM (Supplementary Fig. 4), particularly in the subtropical gyres (Fig. 3, compare a and b). These DCMs coincided with biomass maxima, particularly of small-bodied phytoplankton predated by microzooplankton. Because the

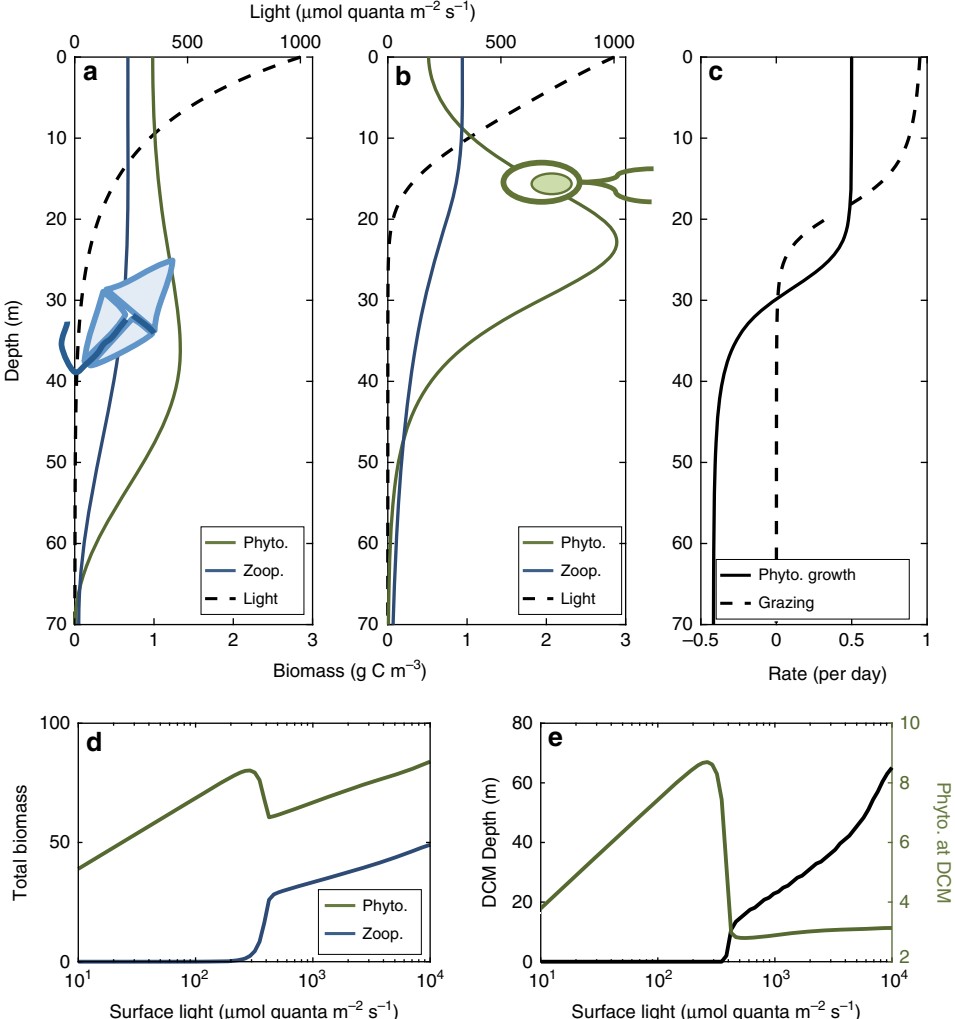

**Fig. 2** Light-dependent grazing drives deep chlorophyll maxima (DCM) formation in a one-dimensional model. **a** In the absence of light dependence of grazing ($H_Z = 0$), phytoplankton are roughly homogeneously distributed where sufficient light is available to support persistence. **b** However, when light dependence is introduced ($H_Z = 50$), a deep phytoplankton biomass maximum corresponding to a deep chlorophyll maximum emerges. **c** Phytoplankton accumulation in a DCM arises from two processes: elevated grazing near the water column's surface, and depressed growth due to light limitation below the compensation depth. **d** Total biomass of both organisms (in g C m$^{-2}$) is a function of surface input light. Once light availability has increased beyond the compensation irradiance for the phytoplankter, total phytoplankton biomass increases with increasing light until light levels are sufficiently high to sustain microzooplankton. At this point, an increase in total microzooplankton biomass slightly suppresses total phytoplankton biomass relative to lower and higher light levels, but at higher light levels the total biomass of both organisms is an increasing function of light. **e** A DCM forms only when both phytoplankton and microzooplankton are present in the water column. The depth of this DCM increases with increasing surface input light, but phytoplankton biomass concentration (g C m$^{-3}$) at the DCM remains approximately constant. Other parameter values are $k_O = 0.001$, $k_P = 0.1$, $k_Z = 0.0005$, $p = 1$, $l = 0.5$, $g = 20$, $e = 0.1$, $m = 0.05$, $H_P = 0.5$, $H_Z = 50$, $H_A = 20$, and $D = 0.05$

unmodified COBALT model systematically underestimated DCM depths, our modification significantly improves the model's agreement with global DCM predictions derived from empirical algorithms linking satellite data with DCM depth (Fig. 3c, Supplementary Fig. 5; two-sample $z$ test: $p < 0.001$, $z$-statistic = 34.227)[21,50]. We were able to achieve these improvements in DCM fit without reducing COBALT's existing ability to accurately represent the distribution of surface chl-$a$ and nitrate (Supplementary Fig. 6).

We also compared the model's output in a region with large discrepancies (the South Pacific Subtropical Gyre, Fig. 3) with in situ depth profiles of chl-$a$ fluorescence (collected by the Sea Education Association Cruise S272, March–April 2017). Although fluorescence is an imperfect proxy for chl-$a$ concentration[3], data from CTD casts on a transect from New Zealand to

Tahiti show that in situ fluorescence maxima offshore are generally deeper than the COBALT model's predictions (Fig. 4). However, incorporation of light-dependent grazing deepens the model's predicted DCM and increases congruence with field data.

In both the modified and unmodified COBALT model, DCM depth varies seasonally: in general, DCMs are deeper in summer (Supplementary Figs. 7–8), which is consistent with prior findings[21] and the prediction that DCMs should deepen with increasing light availability. Interestingly, incorporating light-dependent grazing produces the greatest deepening of the DCM (i.e., largest magnitude increase in depth between unmodified and modified model runs) in late summer (February to April in the Southern hemisphere; July to September in the Northern hemisphere; Supplementary Fig. 9), a time when stratified, low-

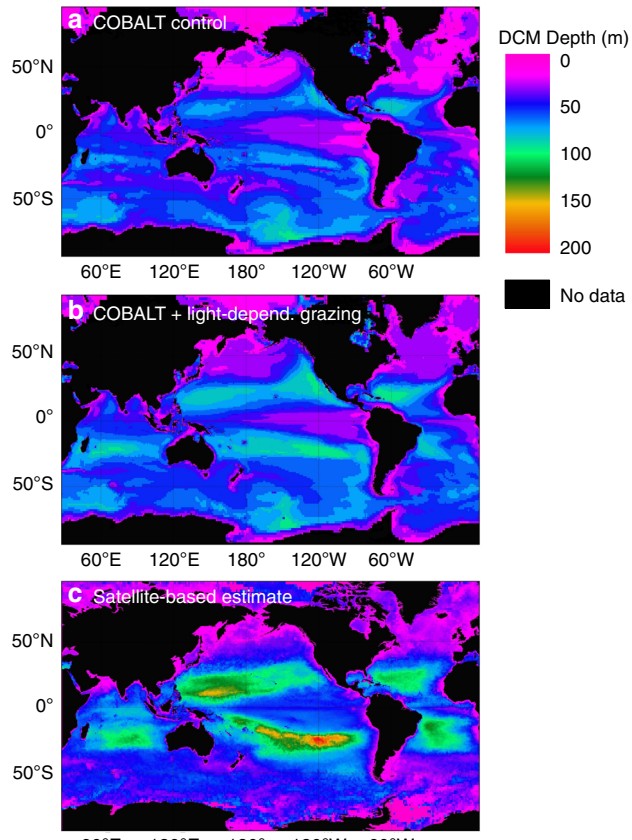

**Fig. 3** Global comparison of deep chlorophyll maxima (DCM) depths predicted by COBALT and satellite algorithms. **a**, **b** Carbon, Ocean Biogeochemistry and Lower Trophics (COBALT) global marine biogeochemistry model output for the unmodified (control) model (**a**) and model modified to incorporate light-dependent grazing of microzooplankton (**b**). Note an overall deepening in the DCM, particularly in the oligotrophic gyres. **c** Empirical estimate of DCM depth using a pigment-based algorithm interpolated using satellite data. Modification of the COBALT model to incorporate light-dependent grazing improves the model's match with these data. All panels are plotted using the same colorbar scale

nutrient systems are most likely to be dominated by grazers[51,52], including mixotrophs[53].

## Discussion

Our findings highlight the importance of top-down controls to regulating the distribution of primary production with depth. Light-dependent grazing is sufficient, in isolation, to drive formation of deep phytoplankton biomass maxima. Evidence for this could be found in water columns where the DCM is decoupled from the nutricline, although, without specific measurements of in situ grazing rates, it is difficult to differentiate between this and other mechanisms (e.g., decoupling of fluorescence and biomass[4], internal waves[54], or sinking detritus[1]). It is likely more common that light-dependent grazing acts alongside other, better-recognized factors to deepen deep biomass maxima. This is evident in our COBALT model runs: even without invoking light-dependent grazing, DCMs form at slightly shallower depths than the nutricline due to nutrient-light co-limitation. Light-dependent grazing deepens these DCMs, but not to the extent that they are also deeper than the nutricline (Supplementary Fig. 10). Thus, light-dependent grazing may be an important

process even in water columns in which the nutricline is deeper than the DCM because the top-down mechanism acts in concert with bottom-up co-limitation. In other words, *even* in water columns where nutriclines are deeper than the DCM, light-dependent grazing can still be an important regulator of DCM location. The deepening induced by light-dependent grazing is particularly pronounced in the subtropical gyres (Fig. 3), which, though relatively unproductive compared to coastal regions, are expected to expand with anthropogenic climate change[55].

Our argument further implies that grazing may limit phytoplankton biomass accumulation in the upper ocean; this is consistent with evidence for high rates of phytoplankton consumption and nutrient recycling in the presence of grazers[29,56,57]. This also implies a fundamental role of grazers in regulating the depth at which biomass accumulates, which may impact carbon export by the biological pump. Though phytoplankton affected by light-dependent grazing are likely to be small-bodied (because they are in stratified water columns), they may nonetheless contribute to export when cells clump and sink[58,59], or when downwelling occurs. Light-dependent grazing may also interact with other, depth-variant grazer traits. For example, elevated temperatures in upper layers of the water column may increase digestion rates[60], further enhancing local grazing rates and compounding the effects of light on grazing. Low light can also limit grazing by larger-bodied zooplankton—for example, visual predators may forage more efficiently in higher light environments[61]—though these impacts are not considered here.

Field studies that quantify membership and activity of the grazer community with depth, or that track the development, depth, and magnitude (phytoplankton biomass) of the DCM over seasonal cycles, can provide tests of this new mechanistic hypothesis for deep chlorophyll maximum formation. Care must be taken to focus such efforts on DCMs that also represent biomass maxima (at least, of specific phytoplankton taxa): our hypothesis predicts variation in abundance of organisms—not just their pigments, which may be further decoupled from fluorescence signals[4]—with depth. While subsurface biomass maxima can be found in all ocean ecosystems and may be composed of a wide variety of phytoplankton taxa[1,62], relatively few studies have systematically measured grazing rates within them or have assessed grazer abundance or composition. Nonetheless, several studies have reported higher concentrations of protist grazers within subsurface chlorophyll maxima[63] and DCMs[64], and mesozooplankton predators are known to vertically migrate in and out of chlorophyll maxima to feed[65,66]. Missing from the rich body of research on DCMs is a systematic study of their grazing communities in relationship to incident irradiance and phytoplankton community composition.

## Methods

**Analysis of published grazing data**. We assembled data from published studies in which grazing data (ingestion rates and, rarely, vacuole clearance rates) were reported for a single grazer offered a single type of phytoplankter prey in controlled laboratory settings. We screened studies to ensure that experimental conditions were identical, except for the manipulation of available light. Where data from multiple prey concentrations were provided, we used the maximum prey density to avoid variation due to Type II functional responses and instead focus on prey-saturated grazing rates. Data were manually extracted from figures and tables, and normalized to a baseline grazing rate by dividing by the grazing rate measured in darkness.

**One-dimensional model**. We formulated our model following Huisman and Weissing[44], except that we introduced a microzooplankter grazer. We follow Huisman and Weissing's formulation in our assumption that light is continuously available (i.e., no day/night light/dark cycles). In order to capture population distributions as a function of depth, we relaxed Huisman and Weissing's assumption of a well-mixed water column and instead allowed organisms to move by diffusion.

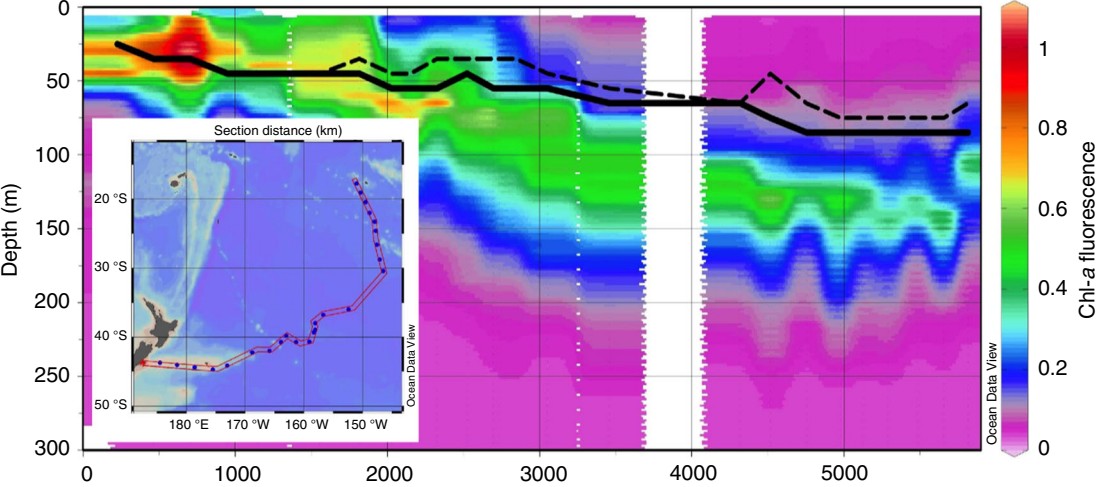

**Fig. 4** Comparison between Carbon, Ocean Biogeochemistry and Lower Trophics (COBALT) model predictions and in situ fluorescence data from the South Pacific Subtropical Gyre. Field data were collected by Sea Education Association (SEA) Voyage S272 (March–April 2017) on a cruise from New Zealand (left side) to Tahiti (right side) (inset). Chlorophyll-a fluorescence data are plotted as a heat map; note a deep deep chlorophyll maxima (DCM) in the oligotrophic basin (right side of the plot). COBALT model output from the same geographic coordinates are overlaid on the plot: The dashed line represents the depth of the DCM in the unmodified COBALT model, and the solid line represents the depth in the COBALT model that incorporates light-dependent grazing. This modification deepened the model's overall prediction of DCM depth, improving its match with in situ data

We simulated the model's behavior using MATLAB (version R2016a, The Mathworks, Inc.). To do this, we discretized the water column into a series of ODEs (ordinary differential equations) representing narrow bins of fixed depth $z'$. We then assumed that each bin was internally well mixed, in order to invoke Huisman and Weissing's formulation for growth of phytoplankton in a well-mixed water column. Photosynthetic growth thus depends on the amount of light entering the top of the bin, $I_i$, and on the amount of biomass within the bin that affects its total absorptivity $\kappa$, where $\kappa$ is given by:

$$\kappa_i = k_0 z' + k_P P_i + k_Z Z_i. \tag{5}$$

We used a centered difference formula to approximate diffusion. Thus, for a particular depth bin $i$, the population dynamics were governed by:

$$\frac{dP_i}{dt} = P_i \left[ \frac{p}{\kappa_i} \log\left( \frac{H_P + I_i}{H_P + I_i e^{-\kappa_i}} \right) - l - \frac{gI_i}{H_Z + I_i} \cdot \frac{Z_i}{H_A + P_i} \right] + D \frac{P_{i+1} + P_{i-1} - 2P_i}{(z')^2}, \tag{6}$$

$$\frac{dZ_i}{dt} = Z_i \left[ \frac{egI_i}{H_Z + I_i} \cdot \frac{P_i}{H_A + P_i} - m \right] + D \frac{Z_{i+1} + Z_{i-1} - 2Z_i}{(z')^2}. \tag{7}$$

Our model water column was closed on top and bottom to flux of biomass; thus the upper and lower depth bins had a modified diffusion approximation that allowed flux only from one bin below or above, respectively.

**Alternative 1-D model formulations.** To test the generality of our results, we formulated three additional water column models. The first of these considered Holling Type I predator functional responses:

$$\frac{\partial P}{\partial t} = P(z) \left[ \frac{pI(z)}{H_P + I(z)} - l - \frac{gI(z)}{H_Z + I(z)} Z(z) \right] + D \frac{\partial^2 P(z)}{\partial z^2}, \tag{8}$$

$$\frac{\partial Z}{\partial t} = Z(z) \left[ \frac{egI(z)}{H_Z + I(z)} P(z) - m \right] + D \frac{\partial^2 Z(z)}{\partial z^2}. \tag{9}$$

The second considered a linear relationship between grazing and light, where microzooplankton can graze at a baseline rate $g_0$, which increases with increasing light at a rate $g$:

$$\breve{g}(I) = g_0 + gI(z). \tag{10}$$

This resulted in the formulation:

$$\frac{\partial P}{\partial t} = P(z) \left[ \frac{pI(z)}{H_P + I(z)} - l - (g_0 + gI(z)) \cdot \frac{Z(z)}{H_A + P(z)} \right] + D \frac{\partial^2 P(z)}{\partial z^2}, \tag{11}$$

$$\frac{\partial Z}{\partial t} = Z(z) \left[ (g_0 + gI(z)) \cdot \frac{eP(z)}{H_A + P(z)} - m \right] + D \frac{\partial^2 Z(z)}{\partial z^2}. \tag{12}$$

The third formulation considered the case in which light availability affected *digestion* rather than attack rates. This formulation parameterizes the mechanism of light-dependent grazing as a changing in handling time of prey cells. More specifically, we imagined that increasing light would reduce handling time according to the function:

$$H_A(I) = \frac{H_A}{I(z)}. \tag{13}$$

This new handling time was incorporated into the Type II grazing functional response:

$$\frac{\partial P}{\partial t} = P(z) \left[ \frac{pI(z)}{H_P + I(z)} - l - \frac{gZ(z)}{1 + gH_A(I)P(z)} \right] + D \frac{\partial^2 P(z)}{\partial z^2}, \tag{14}$$

$$\frac{\partial Z}{\partial t} = Z(z) \left[ \frac{egP(z)}{1 + gH_A(I)P(z)} - m \right] + D \frac{\partial^2 Z(z)}{\partial z^2}. \tag{15}$$

We note that such a formulation has the same qualitative effects as increasing grazing rates in the formulation in the main text of the paper. This is because, in the formulation shown in Eqs. 10–11, in the limit as phytoplankton abundance $P(z)$ approaches infinity, the per-capita zooplankton grazing rate approaches $1/H_A$. Thus, decreasing handling time (by increasing light availability) increases the per-capita grazing rate. This is qualitatively identical to increasing the grazing rate multiplier in Eqs. 2–3.

**COBALT global ocean model.** Details on the formulation of the COBALT model are published elsewhere[46], and our modifications are described in the main text. Here we give a brief description of the components most important for this work, that is, the representation of the plankton community and the parameterization of light limitation.

COBALT represents three phytoplankton types (diatoms, small phytoplankton, and diazotrophs) and three zooplankton types (large, medium, and small zooplankton). Phytoplankton growth is modified by temperature, limited by five nutrients (iron, silicate, phosphorus, nitrate, and ammonia), and limited by light[67]. The irradiance forcing resolves the diurnal cycle and light is attenuated with depth depending on both the attenuation of optically pure seawater and chlorophyll-dependent absorption[68].

Zooplankton feeding is modeled with a Type II Holling functional response. Small zooplankton are assumed to consume bacteria and small phytoplankton, medium zooplankton consume small zooplankton, large phytoplankton, and diazotrophs, and large zooplankton consume medium zooplankton, large phytoplankton, and diazotrophs. Zooplankton feeding is modified by temperature with stronger grazing in warmer water. In the default model version zooplankton feeding is independent of light for all zooplankton types.

**COBALT validation.** To compare our COBALT model runs with existing observational data, we extracted 3-D chl-a distributions (from which we estimated DCM depth), surface nitrate, surface chl-a, and the climatology of the Spring bloom. For each, we computed the percent difference and the root mean square error between each model run and a reference dataset. Results for the first three comparisons are

shown in Supplementary Figs. 5 and 6. No significant differences were observed for Spring bloom climatology, and those data were not high resolution (because model output was saved monthly, rather than daily), so those results are not shown. Our changes to the COBALT model also affected the spatial distribution (i.e., with depth) of phytoplankton and zooplankton; however, available data are too sparse to determine if that improves the representation.

**SEA data collection and analysis**. Field data were collected aboard the SSV *Robert C. Seamans* using a Seabird CTD-mounted Seapoint Chlorophyll Fluorometer (Seapoint Sensors, Inc., Brentwood, NH, USA) and processed using Ocean Data View. Geographic coordinates were recorded at sea using a Furuno GPS Navigator GP-80 type GRP-020 (Nishinomiya, Hyogo, Japan). We used the Quantum Geographic Information Systems[69] Point Sampling Tool to identify the COBALT model's nearest grid cell estimate of DCM depth for each CTD cast.

## Data availability
Grazing data were obtained from peer-reviewed publications (cited in the main text of the manuscript). Global, satellite-based estimates of DCM depth were provided by A. Mignot and are also part of the published literature[21]. Sea Education Association data are available upon written request to the scientific leadership.

## Code availability
Code used in this study to produce the one-dimensional modeling results is provided with this manuscript. COBALT simulation analysis code is available from Charlotte Laufkötter (laufkoetter@climate.unibe.ch) on request.

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

## Acknowledgements

We thank the Sea Education Association and the students and crew of SEA Cruise S272 for collecting and sharing CTD cast data from the South Pacific. We also thank M. Lepori-Bui for assistance in assembling grazing data, A. Mignot for sharing global DCM estimates, J.G. John for providing the COBALT control simulations, E.B. Olson, M.G. Neubert, C.A. Stock, and J.P. Dunne for advice on model formulation, and B.E. Harden for valuable discussion. We thank members of the UCSB EEMB Department for helpful feedback on earlier versions of this manuscript. H.V.M. gratefully acknowledges an NSF Postdoctoral Fellowship (DBI-1401332) and a UBC Biodiversity Center Postdoctoral Fellowship.

## Author contributions

H.V.M. and M.D.J. developed the hypothesis. H.V.M. and C.L. coded and ran model simulations. H.V.M., C.L., and E.M.S. analyzed data and synthesized results, and H.V.M. led writing of the manuscript. All authors contributed to the interpretation of the results and editing of the manuscript.

## Additional information

**Competing interests:** The authors declare no competing interests.

