## [Peer Review File · Nature Communications]

Reviewers' comments:

Reviewer #1 (Remarks to the Author):

The manuscript "Light-Dependent Grazing Can Drive Formation and Deepening of Deep Chlorophyll Maxima" by "Moeller, Laufkötter, Sweeney and Johnson" explores the consequences of light-dependent microzooplankton grazing on the development of a deep chlorophyll maximum (DCM) in the ocean. Light-dependent grazing involves a positive relation between microzooplankton grazing and light. As the authors acknowledge, the mechanism is not new and has been described in laboratory based studies. The authors however, explore whether this mechanism can be responsible for the development of a deep chlorophyll maximum (DCM) in the ocean. They do this in a three-step process. First, they collate data from earlier studies to establish the relation between light and microzooplankton grazing. Second, they develop a simple 1D-model to explore whether the light-dependent grazing alone is sufficient to develop a DCM. Finally, they implement a light-dependent grazing model in a global model (COBALT) and test whether this improves predictions with an appropriate dataset east of New Zealand.

Overall, the manuscript reads well and the topic and findings are novel and of interest for a broad readership. There are however a number of issues that need to be addressed before the manuscript should be considered for publication.

The main issue that I have is the discrepancy between the 1D model formulation and the one used in COBALT. The 1D model has hyperbolic function on light and is linear in phytoplankton biomass, while the COBALT description has a hyperbolic function on phytoplankton biomass and light. While I understand that modifying a complex model like COBALT is difficult, I would expect the 1D model formulation be the same as the one in the COBALT model. Moreover, the data in Fig. 1 suggest that grazing rate is a linear function of light, so it is not clear why the authors used a hyperbolic function to parameterise light dependency in both the 1D and COBALT formulation. Why is there not a clearer relations between the model formulation and the data in Fig. 1?

The mechanisms mentioned on lines 87-96 in fact suggest that the digestion time is reduced due to higher light intensities. The Monod limitation, $f(R)$, can also be written as $f(R) = a \cdot R / (1 + a \cdot h \cdot R)$, in which a is attack rate and h is handling time. The mechanisms described suggest that they decrease handling time h but leave the attach rate a unaffected. In this view, it would make more sense to include light-dependency into the Monod function of phytoplankton grazing.

Although I understand that the mechanisms described favour light-dependent grazing by microzooplankton, but is there also an effect known for mesozooplankton, or has this simply not been studied?

In the 1D model, the biomass of microzooplankton is ~5x higher than phytoplankton biomass. This high grazer:prey ratio enforces the grazing impact, but it is not discussed whether this biomass ratio is supported by field evidence. The data in the paper by Irigoien et al (Journal of Plankton Research 2005) suggests otherwise.

What is the role of temperature in the grazing and digestion process? Temperature will be higher near the surface, which likely also increases digestion rate, so can this act as a co-varying factor that increases microzooplankton grazing closer to the surface?

On lines 169-172 the authors write We therefore tested parameters from a range of possible values to determine a parameter set that lead to a deepening of the DCM while keeping phyto- and zooplankton biomass within the observed range. It is unclear why the authors specifically tune the model to achieve a deepening of the DCM.

Although I agree that the COBALT model with light-dependent grazing improves the fit of the model, this comes at the expense of including additional model parameters so an improved model fit is to be expected. Can the authors apply statistics to show that the model fit increases enough to justify the additional parameters? Moreover, I also wonder what other consequences the inclusion light-dependent grazing has on other COBALT variables. Is there 'only' an improvement of the DCM or do also other variables improve or worsen?

Please also consider the following minor comments:

In 28-30: It is rather trivial to note that the prior mechanisms do not include top-down control, as they are all bottom-up mechanisms.

In 35: "improving the fit of a global model" Improving the fit of what?

In 41: remove numerous

In 51: remove largely (i.e. the mechanisms are only bottom-up)

In 74: please check as opposed the

In 75: I suggest to change wisdom with view

In 81: add year of description of mechanism

In 87: change this pattern with light-dependent grazing

In 104: I suggest to change it seems to have so far been dismissed with has not been thoroughly investigated.

In 118: define "s" in equation 1

In 121-122: remove brackets

In 140: remove However

In 219: reference "Richardson and Jackson (2007) Small Phytoplankton Export from the Surface Ocean Science" seems appropriate here.

Reviewer #2 (Remarks to the Author):

Moeller and colleagues provide an interesting, well written and competent study that suggests a previously unexplored mechanism driving the vertical distribution of chl a fluorescence, indicative of the biomass of primary producers. The ramifications of this work are considerable, because if the authors are correct, then not only does grazing, rather than other factors, drive the vertical distribution of phytoplankton, but more importantly, it implies changes to our understanding of the rates of change in major pools of organic matter as well as their fate in terms of global biogeochemical cycles (e.g. carbon, nitrogen). Using two modeling approaches, the hypotheses are tested numerically, including on a global scale. I really have no concerns about this publication at all and think it is a valid addition to the literature and with the attention that it might command in this short format journal. I do have two food for thought suggestions for the authors:

First, the question whether the DCM represents biomass or fluorescence is addressed, though not resolved. The fact that chl a does not represent phytoplankton biomass let alone reveal biogeochemically important information about species is acknowledged. I think it is a bigger problem than the authors reveal in the 2 sentences in the intro, but they can't be tasked with resolving this. I wish the problem was given more attention (e.g. the biomass at the surface can be much higher than in the DCM (e.g. station Aloha) in which case, the apparent vertical distribution measured is simply a function of the fluorescence per cell due to the vertically varying light field and reduces the significance of detecting the DCM to an instrument issue (e.g. heterogeneity is only observed because of the fluorescence signal, not due to real differences in other properties like Carbon biomass). Graff & Menden-Deuer 2012? explored this in MEPS. It is very clear the authors are aware of this problem and I am not sure what exact consequences my suggestion has, other than that a future empirical test, as rightfully called for, must absolutely go beyond fluorescence measurements.

Second, there are very few studies looking at feeding on vertically heterogeneous phytoplankton distributions in the field. There are a couple using copepods

Dagg, M.J., Frost, B.W. and Walser, W.E., 1989. Copepod diel migration, feeding, and the vertical flux of pheopigments. *Limnology and Oceanography*, 34(6), pp.1062-1071.

> Pierson, JJ, Leising AW, Halsband-Lenk C ... BW Frost. 2005. Vertical distribution and abundance of *Calanus pacificus* and *Pseudocalanus newmani* in relation to chlorophyll a concentrations in Dabob Bay, Washington. *Progress in Oceanography* 67: 349-365.

and a series of papers exploring phytoplankton layer properties, including how population dynamics and predator prey interactions and distributions can drive the intensity and distribution of these layers by Menden-Deuer (2008,2010, 2012) all in MEPS. Those are the only studies I can think of that looked at plankton population dynamics in the field that explore grazing derived drivers of heterogeneity in phytoplankton distributions. Since a natural consequence of your manuscript is to test your hypothesized mechanism at sea, as you call for in your final remarks, I would think it useful to build on and reveal these precedents in the literature.

These two suggestions should not distract from the fact that this is a strong manuscript and very well conceived and prepared. It was a pleasure to review it.

Reviewer #3 (Remarks to the Author):

The paper "Light-Dependent Grazing Can Drive Formation and Deepening of Deep Chlorophyll Maxima" presents the novel hypothesis of light-limited grazing as a driving mechanism for the formation of deep chlorophyll maxima in the global ocean with important implications to modelling systems in use in marine forecasting and Earth System Projections.

While the underlying mechanism is based on sound scientific findings and is clearly presented and in itself plausible, after reading I am not overly convinced of the importance of this process with respect to the traditional paradigm of nutrient-light colimitation.

In order to support their hypothesis, the authors present in a first step a one-dimensional model implementation demonstrating that the process in question on its own is capable of driving the formation of deep chlorophyll maxima. However, the same holds for the classical theory of nutrient-light colimitation.

In a second step the authors present a comparison of a global ocean biogeochemical model system with and without the implementation of light-dependent grazing shown against data of chlorophyll

maxima depths from a satellite model. This comparison shows, that the light-dependent grazing overall deepens the chlorophyll maximum moving the depths obtained closer to those of the satellite model. However, I would argue that the same effect could be obtained via recalibration of the light and nutrient parameters of the original model, while there are no significant structural changes in geographical distribution that would support the hypothesis of a missing processes. Generally, it is hard to judge the significance of the improvement achieved by including this process based on eye-balling the colour maps of a single variable. A more quantitative comparison would help here.

But more importantly, the importance of the mechanism currently missing in standard modelling systems could be more convincingly demonstrated by comparing to additional data that would support the hypothesis. While I appreciate that it is not easy to get hold of coincident vertically distributed microzooplankton and phytoplankton or chlorophyll data at these depths, it should be much easier to involve nutrient data. E.g., in the conclusions the authors state: "Light-dependent grazing is sufficient, in isolation, to drive formation of deep phytoplankton biomass maxima; evidence for this could be found in water columns where the DCM is deeper than the nutricline." Why not supporting this statement with an example case of observational data and the corresponding demonstration of improved modelling capability?

Overall, the paper presents a plausible hypothesis of a potential key driver of deep chlorophyll maxima, but requires more convincing evidence before publication in nature communications.

Additional comments:

Lines 50-80: mixotrophs and their potential role in DCM deserve mentioning in this context (e.g. Tittle et al. 2003).

Line 70: "Yet such mechanisms have been disregarded in DCM formation." That is misleading, it is the light-dependence of the grazing that has been disregarded, not the grazing itself.

Line 198 "Interestingly, incorporating light-dependent grazing produces the greatest deepening of the DCM in late summer (February to April in the Southern hemisphere; July to September in the Northern hemisphere; Supplementary Fig. 5)" To me this seems to be the case for both model versions, not only for the one with light-dependent grazing.

Line 208-211: "This shift is particularly pronounced in the subtropical gyres (Fig. 3), which, though relatively unproductive compared to coastal regions, are expected to expand with anthropogenic climate change." Not entirely clear what shift is intended here, should be phrased better.

Momme Butenschön

Reviewers' comments:

Reviewer #1 (Remarks to the Author):

The manuscript “Light-Dependent Grazing Can Drive Formation and Deepening of Deep Chlorophyll Maxima” by “Moeller, Laufkötter, Sweeney and Johnson” explores the consequences of light-dependent microzooplankton grazing on the development of a deep chlorophyll maximum (DCM) in the ocean. Light-dependent grazing involves a positive relation between microzooplankton grazing and light. As the authors acknowledge, the mechanism is not new and has been described in laboratory based studies. The authors however, explore whether this mechanism can be responsible for the development of a deep chlorophyll maximum (DCM) in the ocean. They do this in a three-step process. First, they collate data from earlier studies to establish the relation between light and microzooplankton grazing. Second, they develop a simple 1D-model to explore whether the light-dependent grazing alone is sufficient to develop a DCM. Finally, they implement a light-dependent grazing model in a global model (COBALT) and test whether this improves predictions with an appropriate dataset east of New Zealand.

Overall, the manuscript reads well and the topic and findings are novel and of interest for a broad readership. There are however a number of issues that need to be addressed before the manuscript should be considered for publication.

Thank you very much – we appreciate your thoughtful and thorough review. We have incorporated your suggestions (which we address individually below) throughout our manuscript.

The main issue that I have is the discrepancy between the 1D model formulation and the one used in COBALT. The 1D model has hyperbolic function on light and is linear in phytoplankton biomass, while the COBALT description has a hyperbolic function on phytoplankton biomass and light. While I understand that modifying a complex model like COBALT is difficult, I would expect the 1D model formulation be the same as the one in the COBALT model. Moreover, the data in Fig. 1 suggest that grazing rate is a linear function of light, so it is not clear why the authors used a hyperbolic function to parameterise light dependency in both the 1D and COBALT formulation. Why is there not a clearer relations between the model formulation and the data in Fig. 1?

Thank you for this suggestion. While originally our choice of the different formulations (Type I in the 1-D model, and Type II in COBALT) was intended to illustrate the generality of the phenomenon, we agree that this is unnecessarily confusing. We have therefore changed the formulation in the main text of the manuscript to be a Type II functional response (consistent with COBALT). Qualitatively, the results of our model remain unchanged (Fig. 2).

Because we consider this demonstration of generality to be important, we have still included the original (Type I) formulation, but we have moved it (as well as other,

new, model formulations including your suggestion of modifying handling time) to the supplementary material (Fig. 3). We note in the main text (lines 161-165):

“These predictions appear to be a general consequence of incorporating light-dependent microzooplankton grazing: other model formulations, including a Holling Type I functional response, a linear (rather than saturating) relationship between light availability and grazing rate, and a formulation in which prey handling time (rather than grazing rate) was a function of light all produced qualitatively identical results (Supplementary Fig. 3).”

Note that in all cases, we have continued to approximate the grazing response to light as a saturating function. In fact, our choice of a saturating relationship between grazing rate and light was grounded in the data: The log-log scale of Figure 1, while necessary to consolidate diverse datasets, also somewhat obscures the exact relationship between light and grazing rate. Of the datasets we considered that had more than two data points, the majority (6 out of 7) showed a non-linear (saturating) relationship between grazing and light in the raw data.

The mechanisms mentioned on lines 87-96 in fact suggest that the digestion time is reduced due to higher light intensities. The Monod limitation, $f(R)$, can also be written as $f(R) = a \cdot R / (1 + a \cdot h \cdot R)$, in which a is attack rate and h is handling time. The mechanisms described suggest that they decrease handling time h but leave the attack rate a unaffected. In this view, it would make more sense to include light-dependency into the Monod function of phytoplankton grazing.

We agree that the proposed (though untested) mechanisms do suggest that the change in overall grazing rate is due to a decrease in handling time with increasing light. However, except for Strom (2001) who also reports clearance rates, the literature demonstrating light-dependent grazing has reported only attack rates. Therefore, it does not seem appropriate at this time to place the main focus of our modification on handling time.

To incorporate your feedback, we have considered an additional formulation of the 1-D model, in which light affects handling time, rather than attack rate; the results are qualitatively identical to those produced by the other 1-D models.

These results are included in the supplementary material (Supp. Fig. 3). This is not surprising because, actually, modifying handling time in the Holling Type II formulation has the same qualitative effect as increasing attack rate in a Monod formulation, as we now describe in the supplementary text (lines 72-77):

“We note that such a formulation has the same qualitative effects as increasing grazing rates in the formulation in the main text of the paper. This is because, in the formulation shown in Eqs. 10-11, in the limit as phytoplankton abundance $P(z)$ approaches infinity, the per capita zooplankton grazing rate approaches $1/H_A$. Thus, decreasing handling time (by increasing light availability) increases the per capita

grazing rate. This is qualitatively identical to increasing the grazing rate multiplier in main text Eqs. 2-3.”

Although I understand that the mechanisms described favour light-dependent grazing by microzooplankton, but is there also an effect known for mesozooplankton, or has this simply not been studied?

Certainly, light availability can affect other zooplankton, for example visual predators that use light to find prey. Unfortunately, it is beyond the scope of this manuscript to consider this phenomenon in detail (257-259):

“Light may also affect grazing by larger-bodied zooplankton—for example, visual predators may forage more efficiently in higher light environments⁵⁷—though these impacts are not considered here.”

In the 1D model, the biomass of microzooplankton is ~5x higher than phytoplankton biomass. This high grazer:prey ratio enforces the grazing impact, but it is not discussed whether this biomass ratio is supported by field evidence. The data in the paper by Irigoien et al (Journal of Plankton Research 2005) suggests otherwise.

We agree. In our modified model formulation (see response above), we have also adjusted the conversion efficiency parameter to achieve a more realistic ratio of phytoplankton to grazer biomass.

What is the role of temperature in the grazing and digestion process? Temperature will be higher near the surface, which likely also increases digestion rate, so can this act as a co-varying factor that increases microzooplankton grazing closer to the surface?

We agree that warmer surface temperatures could further accelerate local grazing rates by the mechanism you propose. Indeed, the COBALT model already incorporates the temperature-dependence of grazing, and this parameterization was not changed in our simulations. Furthermore, temperature within the mixed layer doesn't vary much, while light always attenuates with depth. So temperature can only affect DCM depth if the DCM is below the mixed layer. Because it is beyond the scope of this manuscript to explicitly consider this compounding mechanism, we have addressed temperature briefly in the final section (lines 254-257):

“Light-dependent grazing may also interact with other, depth-variant grazer traits. For example, elevated temperatures in upper layers of the water column may increase digestion rates⁵⁹, further enhancing local grazing rates and compounding the effects of light on grazing.”

On lines 169-172 the authors write “We therefore tested parameters from a range of possible values to determine a parameter set that lead to a deepening of the DCM while keeping phyto- and zooplankton biomass within the observed range.” It is unclear why the authors

specifically tune the model to achieve a deepening of the DCM.

Thank you for pointing that out, we agree that this sentence is phrased confusingly. The reason for our choice of parameters was that, prior to our modification, COBALT systematically underestimated DCM depths, particularly in the oligotrophic gyres. However, the justification for this was not clear in our initial manuscript.

We have tested a range of parameters for the light-dependent grazing. Some have let to very little changes or a minor deepening of the DCM, but others - the ones we present in the paper - let to more significant changes. Some of the tested parameter values that represented very strong light dependence have let to extinction of the plankton community due to too strong grazing pressure. Those runs have been discarded.

The light dependence of the microzooplankton grazing is currently only weakly constrained by the available laboratory observations. The model runs therefore cannot unequivocally confirm that this mechanism is an important effect in the field, but it is well within the range of the experimental uncertainty that they are - which is what we wanted to demonstrate. We have rephrased this as (lines 189-193):

“We therefore tested parameters from a range of possible values within the weak experimental constraints to determine a parameter set that improved the match between the model’s predicted DCM depths and observed data, while keeping phyto- and zooplankton biomass within the observed range.”

Although I agree that the COBALT model with light-dependent grazing improves the fit of the model, this comes at the expense of including additional model parameters so an improved model fit is to be expected. Can the authors apply statistics to show that the model fit increases enough to justify the additional parameters? Moreover, I also wonder what other consequences the inclusion light-dependent grazing has on other COBALT variables. Is there ‘only’ an improvement of the DCM or do also other variables improve or worsen?

We agree that a major challenge in any modeling exercise is balancing improvements to model fit with losses in degrees of freedom due to additional parameters. Thanks to your suggestion (and similar comments from Reviewer 3), we have included a statistical quantification of the improvement of our model’s fit to existing DCM data. In brief, we compared the relative difference between the COBALT model runs and the satellite data, and demonstrated that the modified COBALT model had a statistically significantly better fit (lines 204-207):

“Because the unmodified COBALT model systematically underestimated DCM depths, our modification significantly improves the model’s agreement with global DCM predictions derived from empirical algorithms linking satellite data with DCM depth (Fig. 3C, Supplementary Fig. 5; Two-sample z-test: $p < 0.001$, z-statistic = 34.227)^{20,49}.”

We have also taken your suggestion to consider changes in other variables, and have found negligible changes to the timing of the spring bloom, surface nitrate, or surface chlorophyll-a (Supplement; Supplementary Figure 6).

Philosophically, our modification to the COBALT model is more complex than simply adding three new parameters. In particular, we have incorporated an entirely new mechanism into the model, which has strong justification grounded in first principles (literature evidence for light-dependent grazing). Thus, the statistical tests one might employ to determine the necessity of, for example, explanatory variables in a statistical model, or to compare across models (e.g., AIC), are not appropriate here.

The question how complex a model should be is controversially debated in the scientific community (e.g., Anderson 2005) and there is currently no obvious answer. Adding additional parameters or processes to marine ecosystem models does not necessarily lead to an improved fit, as is evident by many model comparison papers where more complex models don't outperform simpler parameterizations (e.g., Hashioka et al. 2013, Kriest et al. 2010, Laufkötter et al. 2015, Sailley et al. 2013). Yet these models are nonetheless thought to have scientific value because (as we hope we have done here) they demonstrate the possible magnitude and nature of the impact of light-dependent microzooplankton grazing relative to a baseline simulation based on widely used ecosystem parameterizations.

*Anderson, T. R. (2005). Plankton functional type modelling: running before we can walk? *Journal of Plankton Research*, 27(11), 1073–1081.*

<http://doi.org/10.1093/plankt/fbi076>

*Hashioka, T., Vogt, M., Yamanaka, Y., Le Quéré, C., Buitenhuis, E. T., Aita, M. N., et al. (2013). Phytoplankton competition during the spring bloom in four plankton functional type models. *Biogeosciences*, 10(11), 6833–6850.*

<http://doi.org/10.5194/bg-10-6833-2013>

*Kriest, I., Khatiwala, S., & Oschlies, A. (2010). Towards an assessment of simple global marine biogeochemical models of different complexity. *Progress in Oceanography*, 86(3-4), 337–360. <http://doi.org/10.1016/j.pocean.2010.05.002>*

*Laufkötter, C., Vogt, M., Gruber, N., Aita-Noguchi, M., Aumont, O., Bopp, L., et al. (2015). Drivers and uncertainties of future global marine primary production in marine ecosystem models. *Biogeosciences*, 12(23), 6955–6984.*

<http://doi.org/10.5194/bg-12-6955-2015>

*Sailley, S. F., Vogt, M., Doney, S. C., Aita, M. N., Bopp, L., Buitenhuis, E. T., et al. (2013). Comparing food web structures and dynamics across a suite of global marine ecosystem models. *Ecological Modelling*, 261-262, 43–57.*

<http://doi.org/10.1016/j.ecolmodel.2013.04.006>

Please also consider the following minor comments:

In 28-30: It is rather trivial to note that the prior mechanisms do not include top-down control, as they are all bottom-up mechanisms.

It is absolutely true that the reason that the prior mechanisms do not include top-down controls is that they are bottom-up. Still, with respect, we have elected to leave the phrasing in the abstract as it is written. We feel that this sentence is necessary to highlight that the aforementioned mechanisms are bottom-up, and to highlight the gap in current mechanistic approaches to predicting DCM formation: a lack of attention to top-down controls.

ln 35: “improving the fit of a global model” Improving the fit of what?

We have rephrased as (lines 36-37):

“...improving agreement between a global model’s predicted DCM depths with observational data.”

ln 41: remove numerous

Thank you. We have made this change.

ln 51: remove largely (i.e. the mechanisms are only bottom-up)

Although this particular paragraph focuses on bottom-up controls, it is not strictly true that top-down controls have been completely ignored. Indeed, we have expanded the following paragraph to describe one example (in mixotrophy) in which grazing has been implicated in DCM formation. Therefore, we find it appropriate to retain this word.

ln 74: please check as opposed the

Thank you. We have rephrased as “...as opposed to a result...”.

ln 75: I suggest to change wisdom with view

We have made this replacement.

ln 81: add year of description of mechanism

We have added the year (2001).

ln 87: change this pattern with light-dependent grazing

We have revised to, “The mechanism underlying the light-dependence of grazing is not fully known.” (line 97)

ln 104: I suggest to change it seems to have so far been dismissed with has not been thoroughly investigated.

Actually, the language in the cited papers has implied dismissal. We have therefore modified this section of the text to read (lines 113-114):

“While this possibility has occasionally been alluded to in the literature^{1,7}, it seems to have so far been dismissed without thorough investigation.”

ln 118: define “s” in equation 1

Thanks for catching this! We have added a clarifying sentence (lines 128-130):
“Thus the *in situ* light availability at any focal depth z can be found by integrating absorption in the water column above the focal depth (represented by the integral over the spatial coordinate, s):”

ln 121-122: remove brackets

We have done so.

ln 140: remove However

Thank you. We have made this change.

ln 219: reference “Richardson and Jackson (2007) Small Phytoplankton Export from the Surface Ocean Science” seems appropriate here.

Thank you for pointing out this highly appropriate reference, which we have now included.

Reviewer #2 (Remarks to the Author):

Moeller and colleagues provide an interesting, well written and competent study that suggests a previously unexplored mechanism driving the vertical distribution of chl a fluorescence, indicative of the biomass of primary producers. The ramifications of this work are considerable, because if the authors are correct, then not only does grazing, rather than other factors, drive the vertical distribution of phytoplankton, but more importantly, it implies changes to our understanding of the rates of change in major pools of organic matter as well as their fate in terms of global biogeochemical cycles (e.g. carbon, nitrogen). Using two modeling approaches, the hypotheses are tested numerically, including on a global scale. I really have no concerns about this publication at all and think it is a valid addition to the literature and with the attention that it might command in this short format journal. I do have two food for thought suggestions for the authors:

Thank you for your kind words about our manuscript! We appreciate and have incorporated your suggestions, which have certainly strengthened the manuscript.

First, the question whether the DCM represents biomass or fluorescence is addressed, though not resolved. The fact that chl a does not represent phytoplankton biomass let alone reveal biogeochemically important information about species is acknowledged. I think it is a bigger problem than the authors reveal in the 2 sentences in the intro, but they can't be tasked with resolving this. I wish the problem was given more attention (e.g. the biomass at the surface can be much higher than in the DCM (e.g. station Aloha) in which case, the apparent vertical distribution measured is simply a function of the fluorescence per cell due to the vertically varying light field and reduces the significance of detecting the DCM to an instrument issue (e.g. heterogeneity is only observed because of the fluorescence signal, not due to real differences in other properties like Carbon biomass). Graff & Menden-Deuer 2012[?] explored this in MEPS. It is very clear the authors are aware of this problem and I am not sure what exact consequences my suggestion has, other than that a future empirical test, as rightfully called for, must absolutely go beyond fluorescence measurements.

We agree that this is a major challenge in the study of DCMs! We do note that in our COBALT model runs, the DCMs that we observed were also biomass maxima; however, this is certainly not always the case in the field, and, even when the DCM is coincident with the biomass maximum, it is not always the productivity maximum.

We have tried to further emphasize this issue throughout the text, including in lines 231-235:

“Evidence for this could be found in water columns where the DCM is decoupled from the nutricline, although, without specific measurements of in situ grazing rates, it is difficult to differentiate between this and other mechanisms (e.g., decoupling of fluorescence and biomass⁴, internal waves⁵⁴ or sinking detritus¹).”

and, in our concluding call for further field research (lines 264-267):

“Care must be taken to focus such efforts on DCMs that also represent biomass maxima (at least, of specific phytoplankton taxa): Our hypothesis predicts variation in abundance of organisms—not just their pigments, which may be further decoupled from fluorescence signals⁴—with depth.”

Second, there are very few studies looking at feeding on vertically heterogeneous phytoplankton distributions in the field. There are a couple using copepods

> Dagg, M.J., Frost, B.W. and Walser, W.E., 1989. Copepod diel migration, feeding, and the vertical flux of pheopigments. *Limnology and Oceanography*, 34(6), pp.1062-1071.

> Pierson, JJ, Leising AW, Halsband-Lenk C ... BW Frost. 2005. Vertical distribution and abundance of *Calanus pacificus* and *Pseudocalanus newmani* in relation to chlorophyll a concentrations in Dabob Bay, Washington. *Progress in Oceanography* 67: 349-365.

and a series of papers exploring phytoplankton layer properties, including how population dynamics and predator prey interactions and distributions can drive the intensity and distribution of these layers by Menden-Deuer (2008,2010, 2012) all in MEPS. Those are the only studies I can think of that looked at plankton population dynamics in the field that explore grazing derived drivers of heterogeneity in phytoplankton distributions. Since a natural consequence of your manuscript is to test your hypothesized mechanism at sea, as you call for in your final remarks, I would think it useful to build on and reveal these precedents in the literature.

We agree and have incorporated these suggestions in our concluding remarks (lines 267-274):

“While subsurface biomass maxima can be found in all ocean ecosystems and may be composed of a wide variety of phytoplankton taxa^{1,62}, relatively few studies have systematically measured grazing rates within them or have assessed grazer abundance or composition. Nonetheless, several studies have reported higher concentrations of protist grazers within subsurface chlorophyll maxima⁶³ and DCMs⁶⁴, and mesozooplankton predators are known to vertically migrate in and out of chlorophyll maxima to feed^{65,66}. Missing from the rich body of research on DCMs is a systematic study of their grazing communities in relationship to incident irradiance and phytoplankton community composition.”

These two suggestions should not distract from the fact that this is a strong manuscript and very well conceived and prepared. It was a pleasure to review it.

Reviewer #3 (Remarks to the Author):

The paper “Light-Dependent Grazing Can Drive Formation and Deepening of Deep Chlorophyll Maxima” presents the novel hypothesis of light-limited grazing as a driving mechanism for the formation of deep chlorophyll maxima in the global ocean with important implications to modelling systems in use in marine forecasting and Earth System Projections. While the underlying mechanism is based on sound scientific findings and is clearly presented and in itself plausible, after reading I am not overly convinced of the importance of this process with respect to the traditional paradigm of nutrient-light colimitation.

Dear Dr. Butenschön—

*Thank you for your thoughtful review. We agree that the crux of the problem in the field is to ultimately be able to differentiate between multiple hypotheses for DCM formation. When DCMs are also biomass maxima (or, at least, maxima in the biomass of some phytoplankton taxa), the two major hypotheses are nutrient-light co-limitation and light-dependent grazing. However the latter hypothesis has not yet been (to our knowledge) clearly demonstrated nor widely accepted in the literature. Therefore, we consider this manuscript’s task to be **first** (and primarily) establishing the significance of light-dependent grazing as a plausible mechanism, and **second** exploring how this mechanism interacts with nutrient-light co-limitation.*

*In our response to your comments (and attendant revisions to the manuscript), we hope that we have clarified how each component of the manuscript contributes to these two goals. We have also provided additional analyses on improvements to model fit that quantify the value of incorporating light-dependent grazing **alongside** co-limitation.*

In order to support their hypothesis, the authors present in a first step a one-dimensional model implementation demonstrating that the process in question on its own is capable of driving the formation of deep chlorophyll maxima. However, the same holds for the classical theory of nutrient-light colimitation.

*Yes, indeed the predictions are identical to those of nutrient-light co-limitation (and of other DCM mechanisms), and this was the purpose of this first component of the manuscript. We believe that it is critical to demonstrate that, even in a deliberately oversimplified water column, light-dependent grazing is sufficient to drive DCM formation. From a purely theoretical perspective, isolating this mechanism provides a rigorous test of its ability to **generate** (rather than merely enhance) DCM formation. This allows us to then argue for consideration of top-down mechanisms on equal footing with bottom-up mechanisms.*

In a second step the authors present a comparison of a global ocean biogeochemical model system with and without the implementation of light-dependent grazing shown against data of chlorophyll maxima depths from a satellite model. This comparison shows, that the light-dependent grazing overall deepens the chlorophyll maximum moving the depths obtained

closer to those of the satellite model. However, I would argue that the same effect could be obtained via recalibration of the light and nutrient parameters of the original model, while there are no significant structural changes in geographical distribution that would support the hypothesis of a missing processes. Generally, it is hard to judge the significance of the improvement achieved by including this process based on eye-balling the colour maps of a single variable. A more quantitative comparison would help here.

We agree that a major challenge in models is justifying the incorporation of new parameters through improvement of model fit. We also agree that the COBALT simulations do not unequivocally prove that light-dependent grazing is an important process in the field. However, we intend to show that it is a plausible explanation for the difficulties many models have in correctly representing the DCM.

We particularly thank you for your suggestion of quantifying improvement in the COBALT model's fit to existing data. We have evaluated the calibrated model (including the light-dependent grazing) with respect to both DCM depth and onset of the spring bloom, and while including light-dependent grazing deepens the DCM it has little affect on the timing of the spring bloom. This shows that light-dependent grazing can in fact alter the DCM depth without significant structural changes at least in temporal chlorophyll distribution. We agree though that for a more quantitative understanding of the importance of this effect in the field, further laboratory experiments and subsequent model studies will be necessary.

We now include a statistical quantification of the improvement of our model's fit to existing DCM data (lines 204-207):

“Because the unmodified COBALT model systematically underestimated DCM depths, our modification significantly improves the model's agreement with global DCM predictions derived from empirical algorithms linking satellite data with DCM depth (Fig. 3C, Supplementary Fig. 5; Two-sample z-test: $p < 0.001$, z-statistic = 34.227)^{20,49}.”

We further note that, indeed, the predicted changes to DCM were geographically variable. This was originally presented in a monthly climatology in Supplementary Figure 5 (now Supplementary Figure 9); we have now added Supplementary Figure 4 which shows mean changes in DCM depth. In particular, the DCM did not deepen uniformly when light-dependent grazing was incorporated, but actually became 20-30m shallower at high latitudes. Deepening was primarily observed in the oligotrophic gyres.

In regards to the hypothesis that the same effect could be achieved by recalibration of the original model, in fact this could only be done if the light and nutrient parameters were made spatially variable (e.g., took on different sets of values in different oceanic regions). The global (spatially invariant) parameters have already been tuned to maximize the model's fit with existing data (e.g., timing and intensity of the Spring Bloom, spatial distribution of primary production; see Stock et al. for details).

Attempts to adjust the global parameters to deepen the DCM have caused mismatches with these other goals (Stock, pers. comm.). Thus, it would be necessary to allow the parameters to vary spatially to capture the spatial heterogeneity in the degree of model-data mismatch. Because including spatial variation in parameter values would introduce a potentially very large set of new parameters, we argue that the incorporation of light-dependent grazing (which involves the use of only three new, global parameters), is actually a more parsimonious way of achieving the model improvements described above.

But more importantly, the importance of the mechanism currently missing in standard modelling systems could be more convincingly demonstrated by comparing to additional data that would support the hypothesis. While I appreciate that it is not easy to get hold of coincident vertically distributed microzooplankton and phytoplankton or chlorophyll data at these depths, it should be much easier to involve nutrient data. E.g., in the conclusions the authors state: “Light-dependent grazing is sufficient, in isolation, to drive formation of deep phytoplankton biomass maxima; evidence for this could be found in water columns where the DCM is deeper than the nutricline.” Why not supporting this statement with an example case of observational data and the corresponding demonstration of improved modelling capability?

In principle, we agree that DCMs in the absence of (or deeper than) nutriclines would provide good evidence for our hypothesis. However, as we have considered this more deeply since our initial submission, we have ultimately decided that this evidence would be insufficient on its own because of myriad other mechanisms that could cause such DCMs. In other words, eliminating the nutrient co-limitation hypothesis does not prove the light-dependent grazing hypothesis because other phenomena such as photoacclimation, internal waves, sinking dying cells, etc., could also be causing DCM formation in such cases.

We have therefore elected to acknowledge this caveat, and re-focus this portion of the discussion on the interaction between light-dependent grazing and light-nutrient co-limitation. We re-examined our COBALT data and found that, for the most part, our DCMs were shallower than the nutriclines – this was true for both the control model, and the modified model. The implication is that, just because a nutricline is present, that does not mean that light-dependent grazing is unimportant. Indeed, light-dependent grazing can drive substantial (statistically significant) deepening of the DCM, even though the DCM will still be above, or adjacent to, the nutricline.

We describe this in lines 230-243:

“Light-dependent grazing is sufficient, in isolation, to drive formation of deep phytoplankton biomass maxima. Evidence for this could be found in water columns where the DCM is decoupled from the nutricline, although, without specific measurements of *in situ* grazing rates, it is difficult to differentiate between this and other mechanisms (e.g., decoupling of fluorescence and biomass⁴, internal waves⁵⁴ or sinking detritus¹). It is likely more common that light-dependent grazing acts

alongside other, better-recognized factors to deepen deep biomass maxima. This is evident in our COBALT model runs: Even without invoking light-dependent grazing, DCMs form at slightly shallower depths than the nutricline due to nutrient-light co-limitation. Light-dependent grazing deepens these DCMs, but not to the extent that they are also deeper than the nutricline (Supplementary Figure 10). Thus, light-dependent grazing may be an important process even in water columns in which the nutricline is deeper than the DCM because the top-down mechanism acts in concert with bottom-up co-limitation. In other words, *even* in water columns where nutriclines are deeper than the DCM, light-dependent grazing can still be an important regulator of DCM location. ”

Overall, the paper presents a plausible hypothesis of a potential key driver of deep chlorophyll maxima, but requires more convincing evidence before publication in nature communications.

Additional comments:

Lines 50-80: mixotrophs and their potential role in DCM deserve mentioning in this context (e.g. Tittle et al. 2003).

We agree. We have revised the paragraph on top-down controls (lines 67-80) to read:

“However, “top-down” controls (e.g., removal by higher trophic levels) are also known to be important drivers of phytoplankton abundance. For example, grazers can regulate phytoplankton abundance and composition^{28,29}, viruses may drive termination of blooms^{30,31}, and in several Earth System Models grazers can even cause a decrease in future primary production³². Yet such mechanisms have generally been disregarded in DCM formation, perhaps because we expect an individual grazer’s functional traits (e.g., per capita ingestion rate) to be independent of depth. One exception has been the observation that mixotrophs (here defined as organisms that simultaneously engage in phototrophy and phagotrophic heterotrophy) may cause DCM formation when (1) they are weaker competitors for light than their phytoplankton prey and thus accumulate and exert top-down control on prey populations at the surface, and (2) they contain relatively low amounts of chlorophyll themselves, such that the DCM coincides with a peak in phytoplankton prey biomass at depth³³. In this case, however, it is the photosynthetic traits of the organisms, rather than those associated with heterotrophy, that are invoked to explain DCM formation.”

Line 70: “Yet such mechanisms have been disregarded in DCM formation.” That is misleading, it is the light-dependence of the grazing that has been disregarded, not the grazing itself.

We agree that there has some been (surprisingly limited!) attention to grazing, though for the most part microzooplankton grazing has been dismissed. We have rephrased this to read “Yet such mechanisms have generally been disregarded” (lines 70-71), and the paragraph goes on to describe mixotrophy as an exception (quoted above).

Line 198 “Interestingly, incorporating light-dependent grazing produces the greatest deepening of the DCM in late summer (February to April in the Southern hemisphere; July to September in the Northern hemisphere; Supplementary Fig. 5)” To me this seems to be the case for both model versions, not only for the one with light-dependent grazing.

It is certainly the case that both models show this trend. However, in this line, we are referring to the difference between the two (previously Supplementary Fig. 5, now Supplementary Fig. 9). We have clarified this by rephrasing (lines 221-223):

“Interestingly, incorporating light-dependent grazing produces the greatest deepening of the DCM (i.e., largest magnitude increase in depth between unmodified and modified model runs) in late summer.”

Line 208-211: “This shift is particularly pronounced in the subtropical gyres (Fig. 3), which, though relatively unproductive compared to coastal regions, are expected to expand with anthropogenic climate change.” Not entirely clear what shift is intended here, should be phrased better.

We agree. We have replaced “shift” with “deepening induced by light-dependent grazing” (line 244):

Momme Butenschön

Reviewers' Comments:

Reviewer #1:

Remarks to the Author:

I thank the Authors for their work on the revision. My comments have been sufficiently addressed.

Reviewer #2:

Remarks to the Author:

Thank you for the thoughtful incorporation of the reviewer comments. It is a good manuscript and I hope it will find attention in the community.

Reviewer #3:

Remarks to the Author:

I'd like to thank the authors for thoroughly clarifying and addressing the comments received improving this manuscript substantially.

My concerns have been well addressed by the authors, in particular the quantification of the altered model dynamics in the three-dimensional experiment strengthens their case considerably.

A further note on the issue of the calibration of the COBALT simulations: there is no question that the COBALT model has been thoroughly optimised for a range of goals, as demonstrated in the work the authors refer to, balancing the various optimisation tasks as stated in reference to the personal communication with Stock. But therefore, in principle, it is not sufficient in order to demonstrate an improvement of the current model, to show an improvement in the metric that is subject to the calibration of the new process itself, but it should at least not harm the other goals and possibly improve them. Having said that, an entire re-assessment of COBALT is certainly beyond the scope of this work and the authors have indicated low impact on the other validation goals paying attention that phytoplankton and zooplankton biomass remain in the observed ranges and providing statistics on the surface nutrient fields, surface chlorophyll and bloom timing in the revised version.

On the base of these reflections, I am now happy to suggest to accept this manuscript for publication in Nature Communications and am curious to see what impact these findings may have on community production and carbon export from the surface ocean in future work.

Momme Butenschön